# Ensemble Machine Learning for the Prediction and Understanding of the Refractive Index in Chalcogenide Glasses

**DOI:** 10.3390/molecules30081745

**Published:** 2025-04-14

**Authors:** Miruna-Ioana Belciu, Alin Velea

**Affiliations:** 1National Institute of Materials Physics, Atomistilor 405A, 077125 Magurele, Romania; miruna.belciu@infim.ro; 2Faculty of Physics, University of Bucharest, Atomistilor 405, 077125 Magurele, Romania

**Keywords:** chalcogenide glasses, refractive index, ensemble learning, small data

## Abstract

Chalcogenide glasses (ChGs) are a class of amorphous materials presenting remarkable mechanical, optical, and electrical properties, making them promising candidates for advanced photonic and optoelectronic applications. With the increasing integration of artificial intelligence in modern materials design, we are able to systematically select, prepare, and optimize appropriate compositions for desired applications in a manner that was unachievable before. This study employs various machine learning models to reliably predict the refractive index at 20 °C using a small dataset of 541 samples extracted from the SciGlass database. The input for the algorithms consists of a selected set of physico-chemical features computed for the chemical composition of each entry. Additionally, these algorithms served as inner models for an ensemble logistic regression estimator that achieved a superior $R^{2}$ value of 0.8985. SHAP feature analysis of the second-best model, CatBoostRegressor ($R^{2}$ = 0.8920), revealed the importance of elemental density, atomic weight, ground state atomic gap, and fraction of p valence electrons in tuning the value of the refractive index of a chalcogenide compound.

## 1. Introduction

Inorganic vitreous compounds that contain no oxygen and include sulfur, selenium, or tellurium are known as chalcogenide glasses (ChGs). Their peculiar structure, characterized by the predominance of covalent and van der Waals bonds between chalcogens and the constituting elements gives them unique mechanical, optical, and electrical properties. Sulfur and selenium naturally tend to form concatenated structures, either octaatomic molecules or long -S-S-/-Se-Se- randomly tangled chains that interact through dispersion forces [1]. In amorphous Te, the chains are much shorter, and some atoms exhibit 1-fold and 3-fold coordination [2]. The existence of additional elements within the chalcogenide matrix generally increases the average coordination number in the material, forming bridges between chains and creating interlaced 2D and 3D disordered architectures. Specifically, halogens terminate chains; group V elements (such as P, As and Sb) are 3-fold coordinated, extending the glass network in 2D; and group IV elements (such as Si and Ge) introduce tetrahedral structures, resulting in a reticulated 3D glass structure. ChGs present low frequency vibrations enabling them to transmit light with wavelengths up to 20 μm (for telluride glasses). Large refractive indices and nonlinear optical phenomena are unique for this class of materials. Today, ChG lenses and waveguides are primary components in thermal imaging devices and optical sensors [1].

The refractive index ($n_{d}$) is a fundamental bulk optical property in the modern glass industry, and its value must be known with precision. It is defined as the ratio between the speed of light in vacuum and its value in the material’s medium. This property depends on the wavelength of light and the characteristics of the material, which vary with temperature. In practice, the refractive index reported for a material is typically measured for two closely spaced wavelengths as follows: the yellow emission line of sodium (589.3 nm) and that for helium (587.6 nm), with only minor differences in their values [3]. However, since most ChGs are opaque or very poor transmitters for visible radiation, measurements of their refractive index are performed at the wavelengths of interest, typically between 1 μm and 12 μm [1].

Certain physical parameters of constituents in the material influence the value of $n_{d}$. For example, in ionic glasses (such as oxidic glasses, fluoride glasses, and borate and phosphate glasses), the polarizability and electronic density of the constituent anions directly determine the refractive index [3]. To infer significant relationships between the physical and chemical parameters and a material property, modern machine learning approaches are preferable to empirical methods. This has been recently proven for the refractive index in a dataset of general inorganic compounds too [4]. Machine learning, a subdomain of artificial intelligence, employs algorithms capable of identifying patterns in data and making predictions about variables. Supervised learning techniques, in particular, rely on input data to predict a target variable [5].

For ChGs, the refractive index was previously studied among other mechanical, electrical, and thermal properties. Mastelini et al. [6] collected data from the SciGlass database, while Singla et al. [7] incorporated the INTERGLAD database. Their coefficients of determination ($R^{2}$) at testing were 0.87 [6] and 0.916 [7], respectively. These studies employed common machine learning algorithms on datasets where the features represented the chemical composition in atomic percents and included a Shapley additive explanations (SHAP) feature interpretation to reveal their interactions and contributions to the property’s value.

The novelty of our study is based upon the feature representation of the data points, which we collected from SciGlass [8] with our criteria. This approach allows for a more fundamental understanding of the relationship between the refractive index and intrinsic atomic properties. Furthermore, our representation is more general, in the sense that our models can predict compositions containing elements outside the training data, provided that their atomic properties are documented.

## 2. Results and Discussion

### 2.1. Performance of the Models

Table 1 presents the three performance metrics for training, cross-validation, and testing for each optimized model. Among all models, CATR achieves the best performance across all datasets, while ADR yields the least satisfactory results on unseen data. The significant improvement in efficiency for ADR compared to its default version is due to a two-step optimization process as follows: tuning the hyperparameters of the inner decision tree model first, followed by optimizing the hyperparameters of the ensemble adaptive boosting model. With the exception of ADR, all models achieved an $R^{2}$ of at least 0.850 on the test data, which demonstrates both reliability and the benefit of optimization.

To further improve the test scores, we combined optimized models using a linear regression meta-model (LR) [9]. The predicted refractive index from the optimized models served as input for training LR, forming what we refer to as the “base set”. Ideally, predictions should be made on completely unseen data, but due to the limited size of the cross-validation set (87 entries), we included samples from the training set, where the prediction error ≥ x% of the real $n_{d}$ value, with x ∈ {5, 6, 7, 8, 9, 10}. Among the different model combinations and x values tested, the best performance was achieved when using ETR, CATR, RFR, and ADR predictions with x = 8 on a base set of 92 entries (5 from the train set) (Table 2).

### 2.2. Prediction of the Refractive Index of Experimental Samples

To further assess the practical accuracy of our machine learning model, we compared the predictions from CATR, the best performing interpretable model, to experimentally measured refractive indices for 24 ChGs in the Si-Ge-Te system. Although Si, Ge, and Te are elements represented in our dataset, ternary compositions comprising exclusively of these elements are entirely absent, except for a single binary composition (Si_20_Te_80_, n_d_ = 3.2). The samples were thin films prepared via magnetron co-sputtering, with their structural and optical properties characterized in a previous study [10]. The resulting coefficient of determination of (R^2^ = 0.1489) is arguably low. As illustrated in Figure 1, while eleven data points closely approach the experimental values, the remaining thirteen points, corresponding to the samples with measured refractive index greater than 3.0, are significantly underestimated by our model. Several factors are likely to contribute to this discrepancy, most notably the sparse distribution of high refractive index data within our training dataset (as seen in Figure 4). Additionally, since most inputs documented in the SciGlass database are bulk glasses, structural differences arising from the sputtering process used for thin film deposition may further influence the refractive index values. The chemical composition and experimentally measured refractive indices at 1550 nm for these samples are provided in Table A1 of Appendix A.

### 2.3. SHAP Feature Importance and Interpretation

With the most reliable model for predicting the refractive index, we proceeded to analyze the physical interpretation of this optical property using the SHapley Additive exPlanations (SHAP) Python package version 0.41.0 [11]. SHAP provides visualization tools to assess the feature impact on the target variable. The Shapley value, derived from game theory, measures the average contribution of a given parameter to the target variable across all possible prediction scenarios [12]. Since the refractive index is dimensionless, so are its Shapley values. Positive values indicate an increase in $n_{d}$ due to a feature, while negative values signify a decrease. Figure 2 illustrates the ranked importance of the 22 features in the optimized CatBoostRegressor. Each data point from the training set is represented as a dot color coded according to the corresponding feature magnitude. The plot reveals trends in the impact of extreme feature values, with clear separation in the top ranked features, indicating their strong influence on the refractive index.

The top four features influencing the value of $n_{d}$ are related to the elemental densities of a composition. Theoretical density refers to a weight-percentage-modified harmonic mean of the elemental densities (see the first formula in Table A2). According to the SHAP plot in Figure 1, these four features have a clear impact on the refractive index. The larger the density of the constituent elements, the higher the value of the optical property. Additionally, the presence of heavier elements (having larger atomic volumes and weights) appears to increase $n_{d}$. The range of heat formation, together with two electronic features, the mean of the ground state gap and the fraction of p valence electrons, exerts a negative effect on the target. However, the average number of valence electrons influences the refractive index positively. This suggests that materials with a higher mean ground state gap and fewer p valence electrons exhibit less optical interaction, while an increased number of valence electrons contributes to stronger electronic transitions, leading to a higher refractive index.

Figure 3 presents selected SHAP dependence plots from the CatBoostRegressor analysis, emphasizing the structural and electronic characteristics most influential for the refractive index in chalcogenide glasses. The values of the features lie on the *x*-axis, and the Shapley values are on the *y*-axis. On the right side of the plots, an interacting feature is represented with the color code for its magnitude. Figure 3a demonstrates that increasing the mean elemental density significantly enhances refractive index, aligning with the Lenz–Lorentz relation (n^2^ − 1)/(n^2^ + 2) = (4π/3)N_A_R_m_ρ, where ρ is the density, R_m_ is the molecule polarizability, and N_A_ is the Avogadro number [13]. Experimental data from the Ge-Sb-Se and Ge-Sb-S-Se-Te systems with compositions with heavier elements showed elevated refractive indices due to increased density [14,15]. This correlation was further validated in Ge-As-Se and Ge-Sb-Se glasses, where density increased and refractive index increased proportionally with higher As and Sb contents [16,17].

Figure 3b reveals a pronounced inverse correlation between the mean ground state electronic gap and refractive index, consistent with the Moss relation n^4^ × E_g_ = K, where K is a constant [18,19]. Tanaka experimentally verified that narrower bandgaps facilitate electronic transitions at lower energies, enhancing polarizability and the refractive index [20]. Furthermore, optical bandgap measurements in Ge-Sb-S glasses showed that narrower bandgaps corresponded to higher nonlinear refractive indices [21,22], while the THz-IR refractive index correlation reported for As-S and As-Se systems [23] further supports this relationship across different spectral regions.

Figure 3c highlights the direct relationship between the mean atomic volume and refractive index. Borisova established that substituting selenium with larger tellurium atoms in chalcogenide glasses increases the refractive index due to enhanced electronic polarizability [24]. This correlation was further substantiated by Petit et al., who demonstrated that glass compositions with larger mean atomic volumes consistently exhibit higher refractive indices due to the more expansive and easily polarizable electron clouds [25].

Figure 3d confirms that a higher mean number of valence electrons strongly correlates with increased refractive indices. This relationship aligns with studies by Tichy and Ticha, who established empirical correlations between the average coordination number and optical properties in chalcogenide glasses [26]. Kumar et al. provided additional experimental evidence in Ge-Se-In systems, demonstrating that compositions with higher average valence electron counts consistently exhibit elevated refractive indices [27].

Finally, Figure 3e illustrates that increasing the fraction of p-valence electrons negatively impacts the refractive index. While less extensively investigated, this relationship is supported by fundamental quantum mechanical principles regarding orbital characteristics. As established by Pauling and later applied to optical materials by Phillips, p-orbitals exhibit greater directionality and spatial localization compared to s-orbitals or d-orbitals, resulting in reduced polarizability [28,29]. Specifically for chalcogenide glasses, Aniya and Shinkawa developed theoretical models that account for the differential contributions of various orbital types to electronic polarizability, and ultimately, the refractive index [30].

The ranking of features emphasizes that, for ChGs, the mass distribution within the structure, both at room temperature and the melting point, has a greater impact on the refractive index than the electrical parameters do. This aligns well with the structural contrast between ChGs and ionic glasses, where the polarizability of anions and cations primarily decreases or increases the velocity of light in the material [3]. It should be noted that the SHAP feature interpretation remains dependent on the model, so its reliability depends on the accuracy and robustness of the model itself.

## 3. Methods

### 3.1. Data Acquisition

Chalcogenide compositions and their refractive index were extracted from SciGlass [8], which is the largest free and open database of glass compositions and properties of glasses compiled from the literature up to 2019. The dataset of this study consists of composition-property samples that meet specific criteria, constraints, and corrections applied to the raw data. The selection criteria for chemical compositions were as follows: the composition must not contain oxides and must have a non-zero amount of chalcogen. No other elements were excluded from consideration to maximize the number of samples. Since the raw data describes the entries as mixtures of compounds and elements with their respective molar percentages, we converted these into the corresponding elements and their atomic percentages. Entries where the total atomic percentages deviated from 100% to 0.05% were discarded. For compositions within this tolerance, the largest atomic percentage was adjusted by adding or subtracting the difference, accordingly. Duplicate entries were then removed. In cases where duplicated compositions were assigned distinct $n_{d}$ values, extreme values (i.e., beyond the 0.05% and 99.95% percentiles) were discarded, and the median value was assigned as the $n_{d}$ for that composition. This approach aligns with the data cleaning methodology of Mastelini et al. [6]. At this stage, extreme values across the entire $n_{d}$ distribution were also eliminated, each represented by a single composition. Finally, compositions containing Dy, Gd, Pr, Tm, and Yb were excluded due to missing values in the fundamental elemental attributes used for feature generation. The final dataset comprise 541 compositions spanning 32 elements, as follows: Li, S, Ge, Cd, B, Al, Si, Ga, Br, La, Er, F, Mn, Tl, Bi, Na, Zn, Pb, P, Te, As, Sb, Se, Cl, Ag, Cs, In, I, Ba, Cu, Hg, and Sn. The distribution of refractive indices is visualized in Figure 4, and descriptive statistics are presented in Table 3.

The frequency of elemental representation in the dataset compositions is shown in Figure 5. S is the most abundant element, appearing in more than 320 samples. Ge is the second most represented, with 308 entries, followed by Se and As. Zn and Li appear only once. Consequently, machine learning models are expected to generalize better for compositions containing elements that are more frequently represented in the dataset.

Figure 6 illustrates that ternary glasses are the most common in the dataset, followed by quaternary and binary glasses. Six samples contain nine elements, while two are monoelemental (S and Se glasses).

### 3.2. Features and Feature Selection

This study examines the relationship between fundamental atomic properties and the refractive index. Consequently, we generated an initial set of 379 computational physico-chemical features. These features were derived from 44 elemental properties collected from the Mendeleev Python package [31], as well as from other publications on the prediction of metallic glasses [32,33,34,35,36,37], and on the theoretical calculations of the atomic characteristics [38,39,40,41,42,43]. The dataset encompasses a wide range of topological, thermodynamic, and electronic properties of the elements. Examples include the first ionization energy, density at room temperature, Mulliken electronegativity, Pauling electronegativity, electrophilicity, ground state gap, melting temperature, density at the melting temperature, number of s/p/d/f valence electrons, atomic/covalent/van der Waals volume, dipole polarizability, and coordination number. The complete list of properties can be found in Appendix A. For a composition containing the elements $X_{1}$, $X_{2}$, $X_{3}$,…, $X_{n}$ with atomic percentages $x_{1}$, $x_{2}$, $x_{3}$,…, $x_{n}$, a vector of the corresponding values is assigned for each elemental property $p_{1}$, $p_{2}$, $p_{3}$,…, $p_{n}$. Features are constructed by applying mathematical functions to these vectors, or to more property vectors, in more complex cases. The formulas for all these functions, some of which are very complex, are detailed in another study [43]. Table A1 in Appendix A includes the formulas necessary to understand the most important features discussed in Section 2.3.

The dataset comprises 541 collected samples divided as follows: 345 (64%) for training, 87 (16%) for cross-validation, and 109 (20%) were left for testing. A relatively large percentage was allocated to testing to ensure model reliability. It is certain that the 379 candidate features exceed the appropriate dimensionality of predictors for only 345 training entries. Feature standardization was applied in the first preprocessing step. The raw values were normalized based on their distribution (the mean and standard deviation) within the training set, as defined by Equation (1), as follows:(1)$$X^{\prime }=\frac{X-\bar{X_{train}}}{\sigma (X_{train})}$$
where X is a raw feature value, $\bar{X_{train}}$ is the mean value across the 345 training entries, and $\sigma (X_{train})$ is the standard deviation. After transformation, the standardized features in the training set have a mean of 0 and a standard deviation of 1. The first step for feature selection involved the computation of the mutual information regression score (MIR) [44]. MIR is a non-negative number that quantifies in bits the dependency between two variables (X and Y) by calculating the difference between the entropy of X and the conditional entropy of X given Y [45]. The 25, 30, 33, and 35 features with the highest-MIR values were evaluated across default models using six-fold repeated training with seven-fold cross-validation. In each iteration, the training set was randomly partitioned in seven equal subsets, with one subset designated for validation and the other six used for training. Repetition mitigates sampling bias in the splits. The model’s reliability was assessed using the coefficient of determination, $R^{2}$ (see Equation (2)). The sets of 30 and 33 features produced the best results across the eight models evaluated. To optimize dimensionality reduction, we selected the 30 features set. Table 4 presents a comparison of the performance metrics for each case with the values rounded to three decimal places. Model names and abbreviations are detailed in Section 2.3 and in the Abbreviations Section.(2)$$R^{2}=1-\frac{\sum_{1}^{n}\left(y_{i}-\hat{y_{i}}\right)}{\sum_{1}^{n}\left(y_{i}-\bar{y}\right)},\;\bar{y}=\frac{1}{n}\sum_{1}^{n}y_{i}\;\;and\;\;\hat{y_{i}}=the\;prediction\;for\;the\;i^{th}\;entry\;$$

The second and final filter for feature selection was recursive elimination (RFS), which uses model-based feature importance scores. This method iteratively removes the least important features at each training step until optimal performance is achieved [46].

Table 5 compares the $R^{2}$ values for training, cross-validation, and testing across the eight models on the 30-feature set selected via MIR, as well as the refined feature sets obtained through RFS. With the exception of one model, both training and validation performance improved after RFS. The HGBR model lacks an RFS step because it does not provide a feature importance method. The final number of features selected for each model is also included.

### 3.3. Machine Learning Workflow

In this study, we employ eight ensemble boosting models based on regression trees from the sklearn, lightgbm, catboost, and xgboost Python packages. These models include random forest (RFR) [47], gradient boosting (GBR) [48], adaptive boosting (ADR) [49], extremely randomized trees (ETR) [50], hist gradient boosting (HGBR) [51], light gradient boosting machine (LGBMR) [52], categorical boosting (CATR) [53], and extreme gradient boosting (XGBR) [54].

A regression tree is a supervised learning algorithm consisting of a series of if-else decisions that yield a continuous numerical output. It is typically represented as an upside-down tree, starting with a root node, followed by internal nodes, and ending with leaf nodes. At each node, the dataset is split based on the best threshold value of a feature to minimize the variance of the target in each resulting node. The tree depth increases until a stopping criterion is met, such as the minimum number of instances in the leaf nodes [55]. Equation (3) mathematically describes the prediction of a regression tree, as follows [55,56]:(3)$$\hat{y_{i}}=\sum_{k=1}^{M}c_{k}\cdot I(x_{i}\in \;R_{k}),\;I=\left\{1\;(x_{i}\in \;R_{k})\;or\;0\right\}\;$$
where $x_{i}$ represents the set of predictors for the $i^{th}$ instance, *k* is the number of the node $R_{k}$ in the tree, and $c_{k}$ and $R_{k}$ are the parameters determined during training.

Regression trees are suitable for capturing complex nonlinear interactions between features, but they tend to overfit because their structure heavily depends on the dataset. Consequently, to enhance generalization, large groups of trees are aggregated into ensembles. We employ two types of ensembles as follows: bagging ensembles, which predict the mean value of their inner regression trees, and boosting ensembles, where each inner tree is trained to correct the errors of its predecessors [55,57].

The random forest regressor (RFR) and extra trees regressor (ETR) are similar algorithms that aggregate predictions from multiple base trees. In the RFR, each tree is trained on a bootstrap sample (sampling with replacement) of the training data, whereas all trees in the ETR receive the same input data. Another difference is how features are split at the nodes, as follows: the RFR selects the optimal threshold value, while the ETR chooses it randomly. Due to the added randomness, the ETR computes faster [58]. Both models are widely used for machine learning tasks due to their strong predictive performance [57,58].

Adaptive boosting (ADR) consists of shallow trees that are sequentially trained on the same input data, while adjusting the weights in the cost function to prioritize examples that were previously more poorly estimated [49]. Gradient boosting trees (GBR) uses gradient descent to optimize a differentiable loss function to train the trees in a sequence. GBR is the simplest model from this approach [55,59]. The histogram-based gradient boosting model (HGBR) determines optimal split points based on feature histograms rather than continuous sorted feature values, reducing computational costs [59]. Light boosting machine (LightGBM/LGBMR), categorical boosting machine (CatBoost/CATR) and extreme gradient boosting machine (XGBoost/XGBR) are highly efficient models handling large and complex datasets. They primarily differ in how they process data types, their objective and regularization functions, and their tree growth strategies. LightGBM grows trees leaf-wise, leading to deeper and narrower structures, whereas XGBoost and CatBoost expand trees level-wise, creating wider structures. CatBoost uses ordered boosting, where trees are trained on subsets of the data while calculating costs on different data points to mitigate overfitting [60,61,62]. XGBoost provides extensive regularization options [54] and CatBoost utilizes symmetric trees, ensuring that all the nodes at a given level use the same feature–split pair, further reducing overfitting [60].

For feature selection, we initially used the default configurations of these models. After selecting the features (a specific set for each model), we fine-tuned the hyperparameters using seven-fold cross-validation, maximizing $R^{2}\;$ as the objective function with the Optuna Python package [63]. Model performance was assessed using the coefficient of determination, mean absolute error (MAE), and mean squared error (MSE), as defined in Equations (4) and (5). Lower MAE and MSE values indicate better predictive accuracy. Section 2.1 discusses the performance improvements in detail.(4)$$MAE=\frac{\sum_{1}^{n}\left|y_{i}-\hat{y_{i}}\right|}{n}$$(5)$$MSE=\frac{\sum_{1}^{n}{\left(y_{i}-\hat{y_{i}}\right)}^{2}}{n}$$

## 4. Conclusions

In this study, we focused on achieving reliable predictions of the refractive index of chalcogenide glasses using common machine learning algorithms.

Our unique dataset comprises 541 examples extracted and processed from the SciGlass database, covering 32 elements from the periodic table, including halides and transition metals. The expanded set of 379 corresponding candidate features, gathered from the literature of elemental physico-chemical attributes, identifies each entry with a vector of mathematical functions based on the chemical composition and the atomic parameters. This dataset can also serve other supervised learning models (e.g., neural networks) either for training or retraining within transfer learning, particularly for the models trained to predict the refractive index of other types of glasses. Additionally, unsupervised clustering models may be employed to identify meaningful patterns and categories of compositions within our data.

The optimized CatBoostRegressor and the LinearRegression stacking learner achieve high test scores, $R^{2}$ = 0.8920 and $R^{2}$ = 0.8985, respectively, comparable with the existing results in the literature. An experimental verification of the CATR model’s predictive capability using compositions from the Si-Ge-Te ternary system, not included in our original dataset, provided further insight into the reliability of our predictions. The model maintained accurate predictions for refractive index values in the range of approximately 2.1–3.0, but it significantly underestimated values above this range, primarily due to the insufficient number of high refractive index samples within the training dataset. The SHAP feature interpretation of the CatBoostRegressor revealed the significant influence of elemental density, atomic ground state gap, and p valence electrons in a composition on the refractive index. These findings are consistent with previous studies into the factors governing refractive index behavior in inorganic materials.

This research further demonstrates the applicability of machine learning techniques for glass design and prediction. However, the limited number of compositions may restrict the generalization power of the algorithms in experiments necessitating the collection of more data by the chalcogenide glass community.

## Figures and Tables

**Figure 1 molecules-30-01745-f001:**
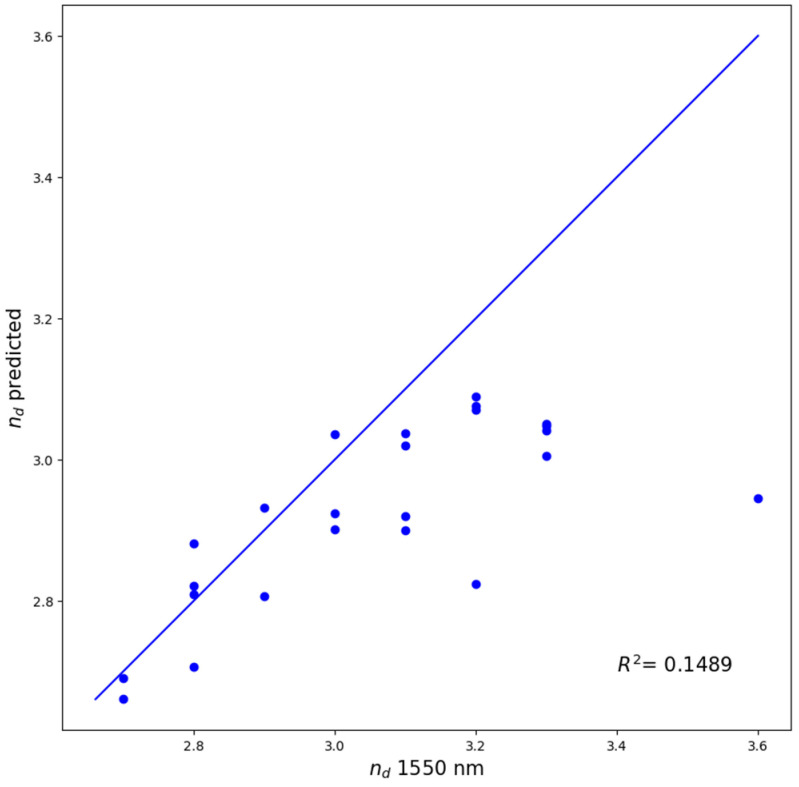
Predictions of the tuned CATR model versus measured values of the refractive index of 24 experimental Si-Ge-Te samples.

**Figure 2 molecules-30-01745-f002:**
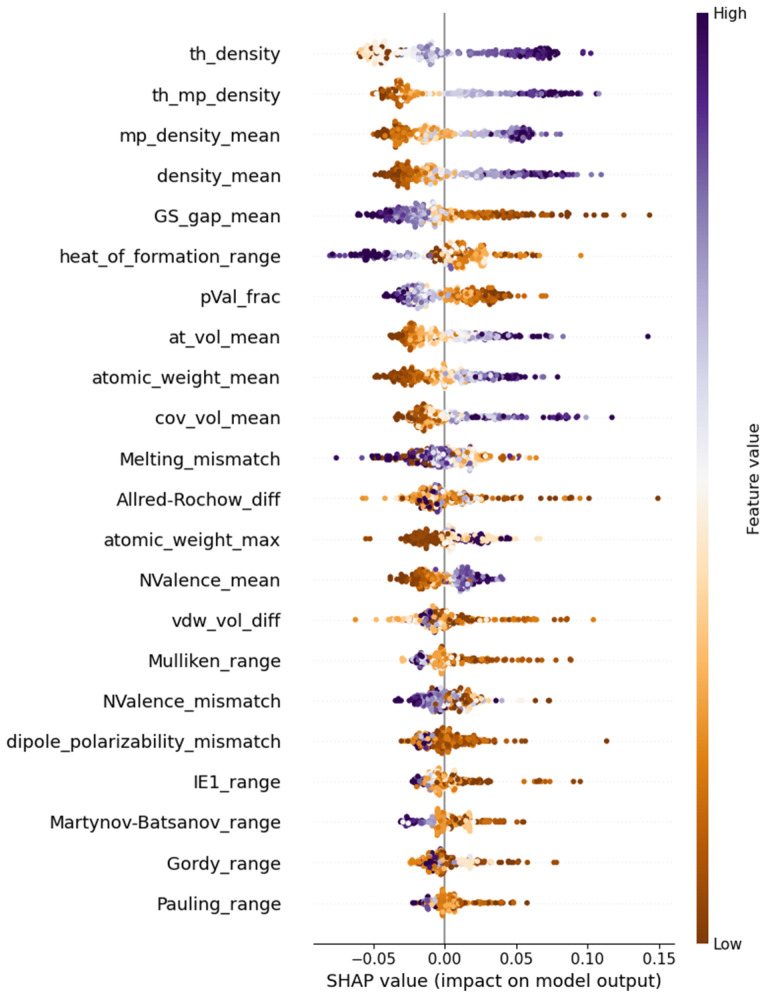
SHAP beeswarm plot of the 22 features in the optimized CATR model.

**Figure 3 molecules-30-01745-f003:**
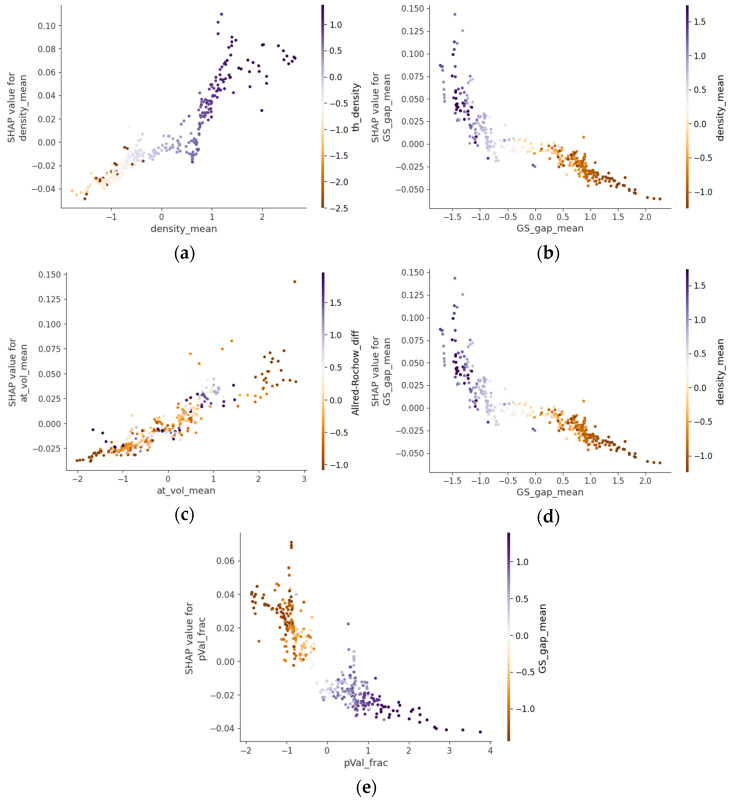
SHAP dependence plots highlighting the structural and electronic features most influential for the refractive index in chalcogenide glasses. The panels depict the relationships of the refractive index SHAP values with the (**a**) mean elemental density, (**b**) mean ground state electronic gap, (**c**) mean atomic volume, (**d**) mean number of valence electrons, and (**e**) fraction of p-valence electrons.

**Figure 4 molecules-30-01745-f004:**
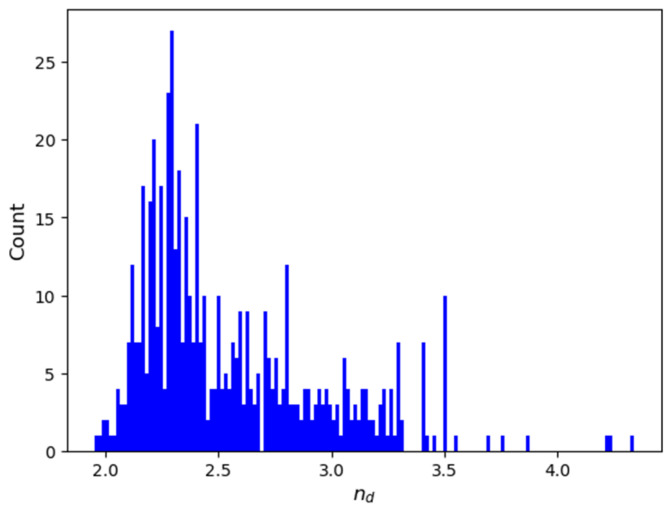
Histogram of the $n_{d}$ distribution after data extraction.

**Figure 5 molecules-30-01745-f005:**
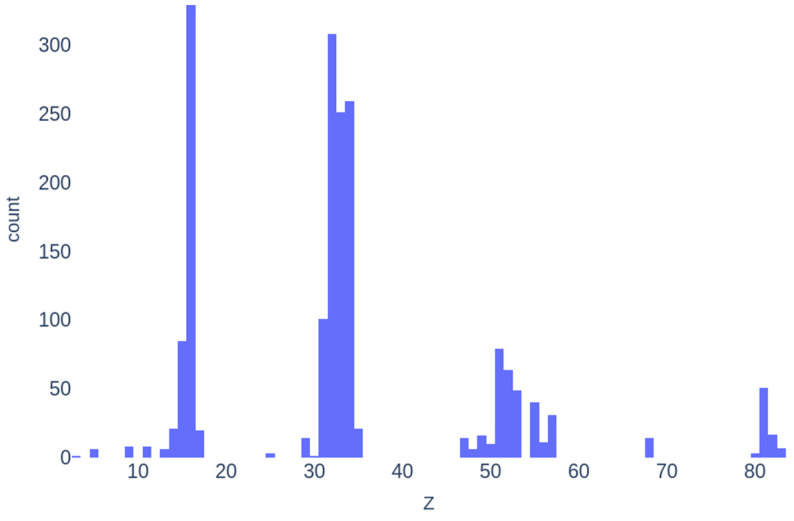
Histogram of atomic number distribution in the dataset.

**Figure 6 molecules-30-01745-f006:**
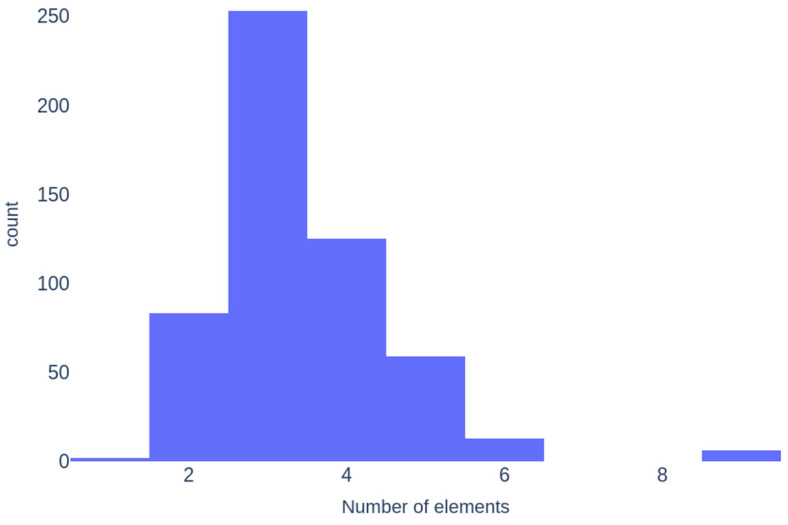
Histogram of the number of elements in compositions.

**Table 1 molecules-30-01745-t001:** Performance of the models after optimization. The highest performance values and the best-performing model are highlighted in bold.

Tuned Model/ Performance	Train Set			Cross-Validation Set			Test Set			Weighted Average Between Test and Cv		
	MAE	MSE	$R^{2}$	MAE	MSE	$R^{2}$	MAE	MSE	$R^{2}$	MAE	MSE	$R^{2}$
RFR	0.047	0.005	0.970	0.081	0.014	0.864	0.097	0.021	0.865	0.090	0.018	0.865
GBR	0.035	0.013	0.925	0.078	0.012	0.879	0.097	0.023	0.856	0.089	0.018	0.866
ADR	0.056	0.004	0.977	0.096	0.020	0.809	0.118	0.028	0.820	0.108	0.024	0.815
ETR	0.004	0.000	1.000	0.072	0.012	0.882	0.089	0.019	0.879	0.081	0.016	0.880
HGBR	0.012	0.004	0.976	0.071	0.010	0.906	0.098	0.024	0.850	0.086	0.018	0.875
LGBMR	0.062	0.009	0.949	0.083	0.015	0.859	0.099	0.022	0.859	0.092	0.019	0.859
**CATR**	**0.004**	**0.000**	**1.000**	**0.069**	**0.010**	**0.904**	**0.084**	**0.017**	**0.892**	**0.077**	**0.014**	**0.897**
XGBR	0.038	0.003	0.982	0.077	0.015	0.854	0.101	0.021	0.868	0.090	0.018	0.862

**Table 2 molecules-30-01745-t002:** Performance of the LR meta-model depending on its base set.

LR’s Input/Metrics	Base Set = Cross-Validation Set			Base Set = Cross-Validation Set + 5 Entries of the Train Set		
	Train Set	Base Set	Test Set	Train Set	Base Set	Test Set
MAE	0.0213	0.0673	0.0826	0.0187	0.0654	0.0810
MSE	0.0009	0.0089	0.0161	0.0005	0.0086	0.0160
R^2^	0.9950	0.9130	0.8976	0.9972	0.9471	0.8985

**Table 3 molecules-30-01745-t003:** Descriptive statistics of the $n_{d}$ values in the dataset.

Minimum	Maximum	Mean	Mode	50% Quantile	Standard Deviation	Kurtosis	Skewness
1.95	4.34	2.55	2.30	2.40	0.40	1.31	1.19

**Table 4 molecules-30-01745-t004:** Performance of the models during training with cross-validation for different feature sets. The highest performance values obtained for each model are highlighted in bold.

$\mathrm{Model}/\bar{R^{2}}\pm \;\sigma (R^{2})$	25 Features	30 Features	33 Features	35 Features
RFR	0.811 ± 0.065	**0.818 ± 0.060**	0.814 ± 0.060	0.814 ± 0.060
GBR	0.779 ± 0.068	**0.800 ± 0.061**	0.783 ± 0.074	0.795 ± 0.067
ADR	0.749 ± 0.081	0.757 ± 0.077	0.754 ± 0.081	**0.762 ± 0.078**
ETR	0.813 ± 0.069	0.812 ± 0.074	0.819 ± 0.072	**0.821 ± 0.071**
HGBR	0.807 ± 0.056	0.815 ± 0.053	0.813 ± 0.058	**0.818 ± 0.054**
LGBMR	0.802 ± 0.059	**0.818 ± 0.055**	0.812 ± 0.060	0.818 ± 0.058
CATR	0.816 ± 0.067	0.833 ± 0.062	0.829 ± 0.066	**0.835 ± 0.067**
XGBR	0.804 ± 0.078	**0.816 ± 0.064**	0.799 ± 0.094	0.810 ± 0.069

**Table 5 molecules-30-01745-t005:** Performance of models with and without RFS. Arrows (↗/↘) and equal signs (=) indicate performance improvements, declines, or no change, respectively, upon applying the additional RFS feature-selection step.

**Model/** $R^{2}$	30 Features Selected with MIR				RFS as Additional Filter			
	Train Set	Cross- Validation (Cv) Set	Test Set	Weighted Average Between Test and Cv	Train Set	Cross- Validation (cv) Set	Test Set	Weighted Average Between Test and cv| Final Number of Selected Features
RFR	0.971	0.869	0.831	0.848	0.979 ↗	0.838	0.868	0.855 ↗ | 28
GBR	0.974	0.861	0.823	0.840	0.976 ↗	0.845	0.848	0.847 ↗ | 23
ADR	0.883	0.794	0.644	0.711	0.899 ↗	0.741	0.731	0.735 ↗ | 17
ETR	1.000	0.879	0.864	0.871	1.000 =	0.868	0.874	0.871 = | 28
HGBR	0.976	0.885	0.822	0.850	-	-	-	-| 30
LGBMR	0.974	0.866	0.783	0.820	0.977 ↗	0.851	0.816	0.832 ↗ | 30
CATR	0.996	0.890	0.862	0.874	0.996 =	0.871	0.886	0.879 ↗ | 22
XGBR	1.000	0.847	0.828	0.836	1.000 =	0.807	0.849	0.830 ↘ | 30

## Data Availability

The data presented in this study are available upon reasonable request from the corresponding author.

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
