# Peer review of "Ensemble Machine Learning for the Prediction and Understanding of the Refractive Index in Chalcogenide Glasses"

_molecules, 2025, doi:10.3390/molecules30081745_

Round 1
Reviewer 1 Report
Comments and Suggestions for Authors
This manuscript is well written and could be appropriate to be published in this joural.
Minor revisions:
1)SHAP analysis lacks connections to ChG structural properties and electronic behavior. More analysis should be added.
2) For conclusion. More actionable insights for glass design based on this work had better be added.
Author Response
Thank you for your comments. Responses are attached.

Reviewer 2 Report
Comments and Suggestions for Authors
This paper discusses the predication of refractive index in Chalcogenide glasses by using machine learning method. This work is scientific sound with encouraging results. It deserves to be published. However, I still have the following questions for the authors.
- Can you try to predict the refractive index of a real material? You may compare the predicted value with measured value to see the accuracy of the prediction.
- What is the wavelength for the predicted refractive index? Is it possible to predict wavelength dependency?
Author Response

(The authors gave the same response as above.)
